# Differences in emergency hospitalization trauma patients during and after the COVID-19 pandemic

**Haifeng Chang** [1]*, **Siyuan Li**[1], **Xijie Ke**[1], **Zhenyu Zhou**[1], **Lijun Zhang**[1], **Baisong Yang**[1], **Bilei Ji**[1], **Liming Jiang**[1], **Yang Yang**[2], **Ting Huang** [1], **Gengwei Zhang** [1]*

1 Department of Emergency Medicine, Shenzhen Third People's Hospital, Second Hospital Affiliated to Southern University of Science and Technology, Shenzhen, China, 2 Shenzhen Key Laboratory of Pathogen and Immunity, National Clinical Research Center for Infectious Disease, State Key Discipline of Infectious Disease, Shenzhen Third People's Hospital, Second Hospital Affiliated to Southern University of Science and Technology, Shenzhen, China

* liuxing@szsdsrmyy10.wecom.work (HC); gengwei@szsdsrmyy10.wecom.work (GZ)

## Abstract

### Background

The spread of COVID-19 has a huge impact on the medical system, affecting the normal order of routine patients, especially obvious effect on the Shenzhen Third People's Hospital as the designated hospital for COVID-19 patients. After the epidemic was loosened in early December 2022, the normal medical order gradually restored in China. How much was the impact on the admission and treatment of emergency trauma patients during and after the epidemic? This study aims to compare the differences between trauma patients admitted to the emergency department during and after the COVID-19 pandemic.

### Methods and findings

The study included all trauma patients admitted through the emergency department from January 2020 to March 2024. Clinical data were collected, and a retrospective comparative analysis was performed on the characteristics of the two groups of trauma patients during and after the pandemic, including gender, age, average length of hospital stay, proportions of admissions to various departments, proportions of severe cases, proportions of surgical patients, and mortality rates, using statistical methods for analysis.

The proportion of male patients during the pandemic (69.98%) was higher than after the pandemic (67.01%). The proportion of patients under 60 years of age during the pandemic was higher than that after the pandemic. There was no significant difference in the average hospital stay between patients during and after the pandemic (P>0.05). Comparisons between the two groups in terms of admissions to departments such as otolaryngology, hepatobiliary surgery, hepatic surgery, orthopedics, urology, neurosurgery, gastroenterology surgery, thoracic surgery, ophthalmology and intensive care unit showed no significant differences (P>0.05). The proportion of surgical patients during the pandemic (75.09%) was

**Data Availability Statement:** All relevant data are within the manuscript and its Supporting information files.

**Funding:** The author(s) received no specific funding for this work.

**Competing interests:** The authors declare no potential conflict of interests. The authors declare that they have no known competing financial interests or personal relationships that could have appeared to influence the work reported in this paper.

higher than after the pandemic (69.53%). The mortality rate during the pandemic (0.13%) was lower than after the pandemic (2.45%).

## Conclusion

The COVID-19 pandemic has impacted trauma patients admitted through the emergency department, with increases in the proportion of male and younger patients, surgical cases, and a decrease in mortality rates during the pandemic.

## Introduction

Since 2019, coronavirus disease 2019 (COVID-19), caused by the severe acute respiratory syndrome coronavirus 2 (SARS-CoV-2), has spread globally [1]. Characterized by rapid transmission, prolonged duration, and high mortality rates of SARS-CoV-2, COVID-19 has significantly threatened human health and economic development. The Third People's Hospital of Shenzhen (National Clinical Research Center for Infectious Diseases) is the designated facility in Shenzhen for treating COVID-19 patients, responsible for the treatment of cases from Shenzhen and its surrounding areas. Since the outbreak, our hospital had also been treating COVID-19 patients alongside routine epidemic prevention and control. This study investigated the impact of the COVID-19 pandemic on the admission of trauma patients to the emergency department over three years.

## Methods

### Data sources

This study was approved by the Ethics Committee of Shenzhen Third People's Hospital. On April 15, 2024, we asked engineers to help retrieve some patient information while protecting patient privacy and personal identity. It included 3,761 trauma patients admitted through the emergency department from January 2020 to November 2022, comprising 2,632 males and 1,129 females (Fig 1A), this group was named the epidemic period group. Additionally, from December 2022 to March 2024, 1,631 trauma patients were admitted, including 1,093 males and 538 females (Fig 2A), this group was named post-epidemic group.

The study included all trauma patients admitted to hospital departments (otolaryngology, hepatobiliary surgery, hepatic surgery, orthopedics, urology, neurosurgery, gastroenterology surgery, thoracic surgery, ophthalmology and intensive care unit) from January 2020 to March 2024 (Figs 1B and 2B).

### Study design

The clinical data were collected for all selected cases, including general information, admission and discharge dates, whether surgical treatment was performed, the department of admission, and the number of deaths. The length of hospital stay was determined according to the time of admission and departure(Figs 1C and 2C). A retrospective comparative analysis was conducted to examine the differences in gender, age,average length of hospital stay, departmental admissions, proportion of severe cases, surgical cases, and mortality rates between trauma patients hospitalized during and after the COVID-19 pandemic. Both groups were divided into four age groups(0–20 years old,20–40 years old,40–60 years old,over 60 years old) (Figs 1D and 2D).

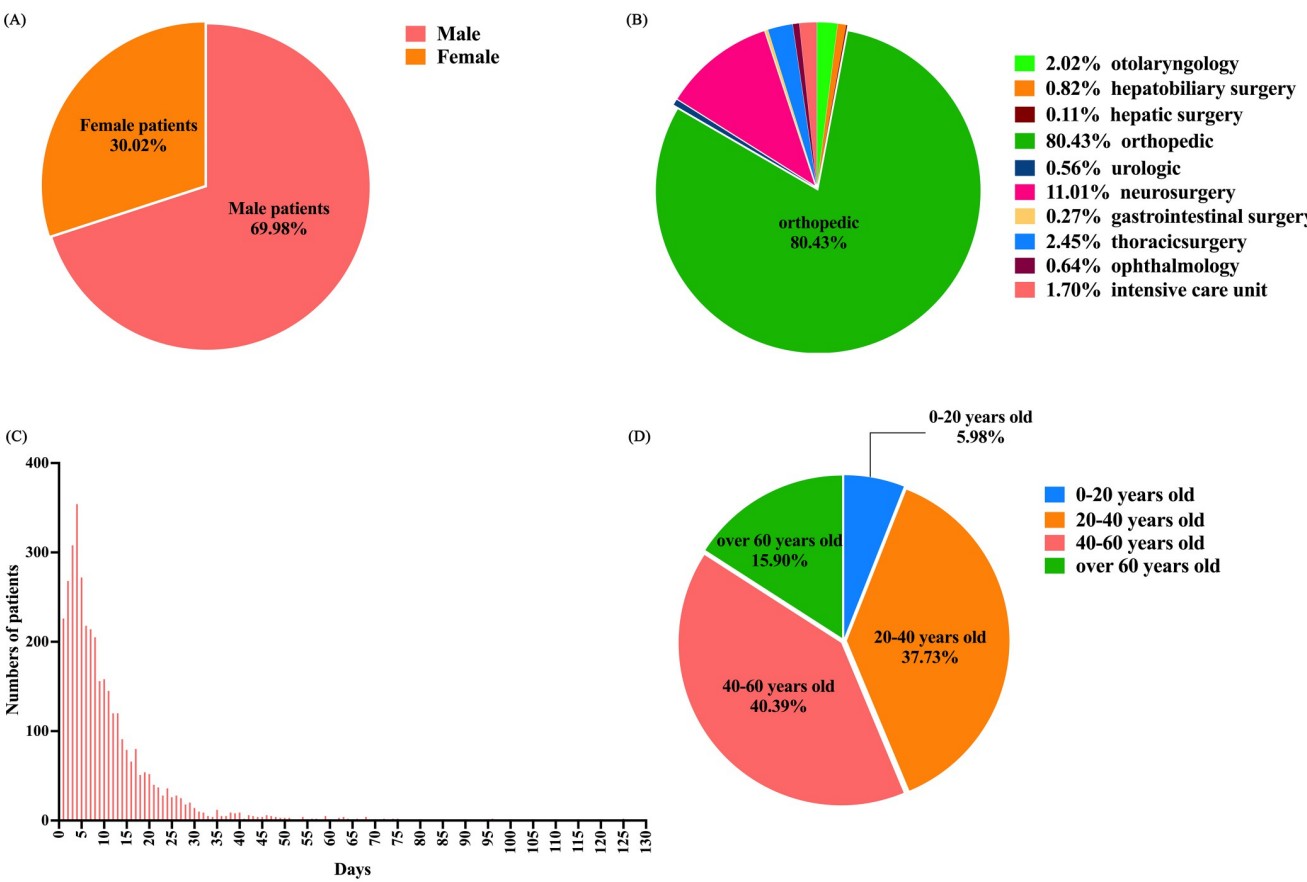

**Fig 1.** A Ratio of males to females in the epidemic period group. This figure shows the gender ratio of trauma patients admitted through the emergency department during the pandemic. B Ratio of patients in different departments in the epidemic period group. This figure shows the ratio of trauma patients admitted through the emergency department to different departments during the pandemic. C The length of hospital stay in the epidemic period group. This figure shows t the frequency distribution of hospital stay duration for trauma patients admitted through the emergency department during the pandemic. D Ratio of four age groups in the epidemic period group. This figure shows the age distribution of trauma patients admitted through the emergency department during the pandemic.

## Statistical analyses

The continuous variables were tested for normality. Normally distributed data were presented as $\bar{x} \pm s$ and compared using independent sample t-tests; non-normally distributed data were expressed as *M (Q1, Q3)* and compared using the Mann-Whitney U test. Categorical data were represented by frequencies and percentages and analyzed using chi-square tests or Fisher's exact tests. All statistical analyses were performed using SPSS 26.0 software. A two-tailed test with $P<0.05$ was considered statistically significant.

## Results

### Comparison of general data

During the pandemic, 2,632 male patients accounted for 69.98% of the study subjects, a proportion significantly higher than the post-pandemic group (67.01%), and the difference was statistically significant ($P<0.05$). The age distribution showed 225 patients aged 0–20 years (5.98%), 1,419 patients aged 20–40 years (37.73%), 1,519 patients aged 40–60 years (40.39%), and 598 patients over 60 years (15.90%). The proportion of patients under 60 years during the

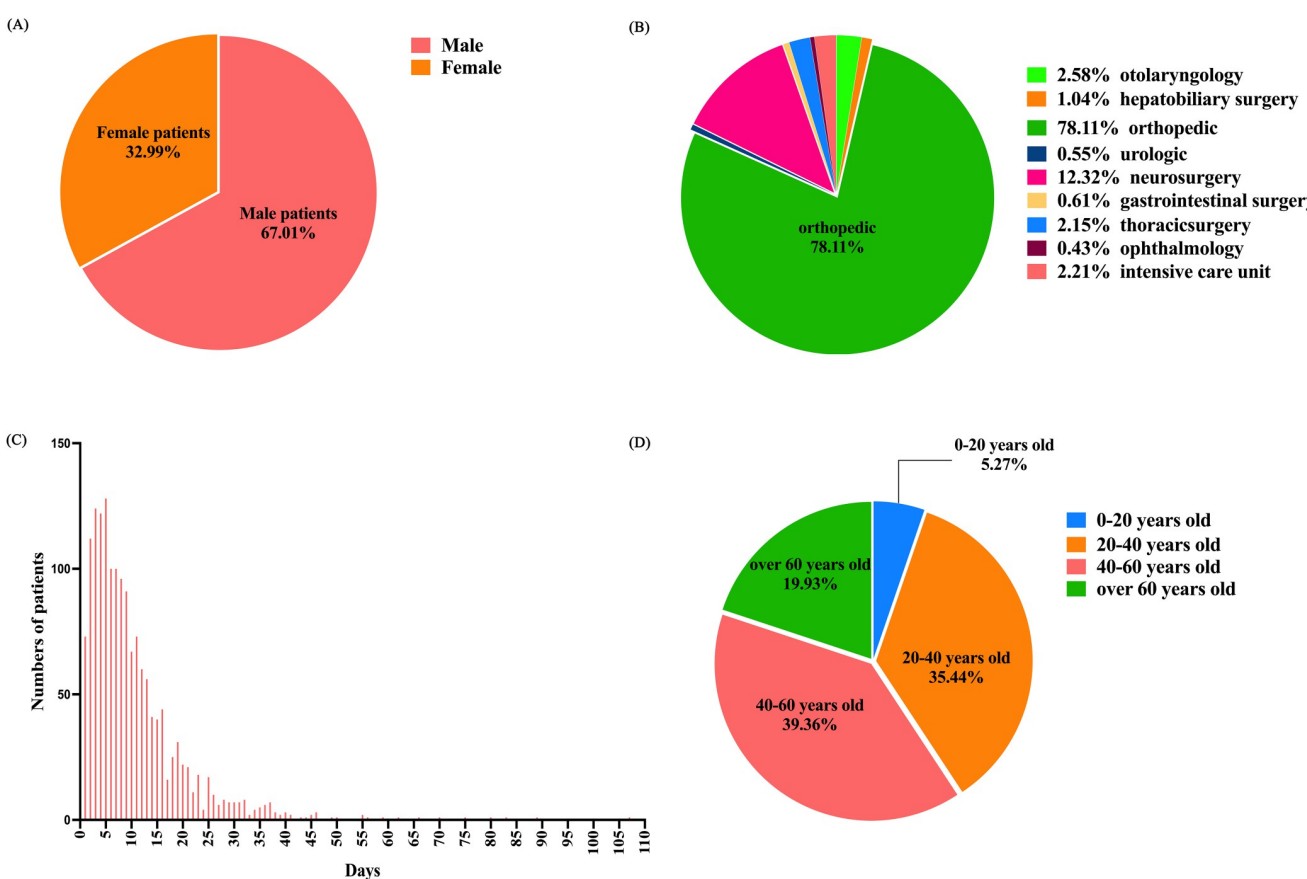

**Fig 2.** A Ratio of males to females in the post-epidemic group. This figure shows the gender ratio of trauma patients admitted through the emergency department after the pandemic. B Ratio of patients in different departments in the post-epidemic group. This figure shows the ratio of trauma patients admitted through the emergency department to different departments after the pandemic. C The length of hospital stay in the post-epidemic group. This figure shows the frequency distribution of hospital stay duration for trauma patients admitted through the emergency department after the pandemic. D Ratio of four age groups in the post-epidemic group. This figure shows the age distribution of trauma patients admitted through the emergency department after the pandemic.

pandemic was significantly higher than that in the post-pandemic group, while those over 60 years were significantly higher in the post-pandemic group($P<0.05$). The average length of hospital stays showed no significant difference between the groups ($P>0.05$; Table 1).

## Admission departments

There were no significant differences in the proportions of patients admitted to departments such as otolaryngology, hepatobiliary surgery, hepatic surgery, orthopedics, urology, neurosurgery, gastroenterology surgery, thoracic surgery, ophthalmology and intensive care unit between the groups during and after the pandemic ($P>0.05$; Table 2).

## Surgery conditions

The proportion of surgical patients during the pandemic was 75.09%, significantly higher than the post-pandemic group (69.53%; $P<0.05$; Table 1).

**Table 1. General data comparison.**

| | the epidemic period group(n = 3761) | post-epidemic group(n = 1631) | statistic | P-value |
|---|---|---|---|---|
| gender, n(%) | | | 4.69 | 0.030 |
| male patients | 2632 (69.98) | 1093 (67.01) | | |
| female patients | 1129 (30.02) | 538 (32.99) | | |
| age, n(%) | | | 13.68 | 0.003 |
| 0–20 years old | 225 (5.98) | 86 (5.27) | | |
| 20–40 years old | 1419 (37.73) | 578 (35.44) | | |
| 40–60 years old | 1519 (40.39) | 642 (39.36) | | |
| over 60 years old | 598 (15.90) | 325 (19.93) | | |
| average length of hospital stay,day | 7 (4–13) | 8 (4–13) | 3.73 | 0.053 |
| proportion of surgical patients, n(%) | 2824 (75.09) | 1134 (69.53) | 18.005 | <0.001 |
| the mortality rate, n(%) | 5 (0.13) | 40 (2.45) | 73.959 | <0.001 |

## Death situation

The mortality rate among study subjects during the pandemic was 0.13%, significantly lower than in the post-pandemic group (2.45%), and the difference was statistically significant (P<0.05; Table 1).

## Discussion

During the pandemic, Shenzhen experienced a rise in unemployment rates and a decrease in population. In our study, the proportion of male trauma patients admitted during the pandemic was higher than in the post-pandemic period. This suggests a decrease in the proportion of female trauma patients admitted during the pandemic (Figs 1A and 2A), potentially linked to an increase in risk factors associated with depressive symptoms caused by COVID-19 epidemic [2], which are generally more prevalent among women [3]. This could have led to reduced activity among women in Shenzhen or a greater outflow of the female population from the city. Our study also found that the proportion of trauma patients under 60 admitted during the pandemic exceeded that of the post-pandemic period (Figs 1D and 2D). Conversely, studies have found fewer depression cases among those over 60 [4]. Shenzhen is a youthful city, with fewer elderly residents, most of whom live with their working-age children. During the pandemic, the increased pressures of life in Shenzhen may have prompted older adults to return to their hometowns. Additionally, increased home stays and reduced outings

**Table 2. Comparison of admissions by department.**

| department, n(%) | the epidemic period group(n = 3761) | the post-epidemic group(n = 1631) | statistic | P-value |
|---|---|---|---|---|
| otolaryngology | 76 (2.02) | 42 (2.58) | 1.633 | 0.201 |
| hepatobiliary surgery | 31 (0.82) | 17 (1.04) | 0.613 | 0.434 |
| hepatic surgery | 4 (0.11) | 0 (0.00) | Fisher | 0.188 |
| orthopedic | 3025 (80.43) | 1274 (78.11) | 3.786 | 0.052 |
| urologic | 21 (0.56) | 9 (0.55) | 0.001 | 0.976 |
| neurosurgery | 414 (11.01) | 201 (12.32) | 1.950 | 0.163 |
| gastrointestinal surgery | 10 (0.27) | 10 (0.61) | 3.712 | 0.054 |
| thoracic surgery | 92 (2.45) | 35 (2.15) | 0.446 | 0.504 |
| ophthalmology | 24 (0.64) | 7 (0.43) | 0.869 | 0.351 |
| intensive care unit | 64 (1.70) | 36 (2.21) | 1.598 | 0.206 |

among the elderly during the pandemic might have led to a decrease in the proportion of elderly trauma patients admitted to hospitals in Shenzhen.

The duration of hospital stays (Figs 1C and 2C) for patients is influenced by factors such as age, gender, mechanism of injury, infection, type of injury, survival rates, and injury severity score (ISS) [5]. During the pandemic, studies in orthopedics revealed an increase in pre-surgical wait times, yet this did not extend the hospital stays for patients with hip fractures [6]. Furthermore, during the pandemic, patients requiring emergency surgery were operated on as swiftly as possible under effective protection. Comparisons of hospital stay durations during and after the pandemic showed no significant changes, indicating that the pandemic did not affect the hospital stay durations for trauma patients in our hospital, which continued to depend on individual patient circumstances and medical conditions.

The research found no significant changes in the proportion of patients admitted to various departments (otolaryngology, hepatobiliary surgery, hepatic surgery, orthopedics, urology, neurosurgery, gastroenterology surgery, thoracic surgery, ophthalmology and intensive care unit) during and after the pandemic (Figs 1B and 2B). This indicates that the admission standards for each department in our hospital were not affected by the spread of the pandemic. Studies such as Maryam Baradaran-Binazir's retrospective analysis of data from Iran's National Trauma Registry during the pandemic showed a reduction in the total number of trauma patients admitted to Intensive care unit and the length of stay at the hospital [7]. Similarly, Concetto Battiato's study on trauma patients in Italy found a notable decrease in trauma cases during the pandemic [8]. Nirupama Kannikeswaran et al. [9] conducted a study on pediatric trauma epidemiology during the early and later stages of the pandemic. The study revealed higher hospitalization numbers in the early phase but a return to pre-pandemic levels in the later stage, which was likely to be influenced by local government regulations and healthcare standards, etc.

Research on the proportion of surgeries among trauma inpatients during and post-COVID-19 pandemic is scant both domestically and internationally. This study observed a higher surgical rate among inpatients during the pandemic. This may be attributed to efforts to minimize human contact during the pandemic, as trauma patients who did not require surgical intervention were treated conservatively, not admitted, and followed up on an outpatient basis. In contrast, the mortality rate among post-pandemic inpatients was higher than that during the pandemic, although the proportion of critically ill patients did not differ significantly between the two periods. There is a paucity of research on whether the mortality rates of patients post-pandemic differ from those during the pandemic. This may be due to the low incidence of patient deaths, which requires further investigation to determine specific causes.

## Supporting information

**S1 Materials. Original data.**
(XLSX)

**S2 Materials. The epidemic period group.**
(XLSX)

**S3 Materials. Post-epidemic group.**
(XLSX)

## Acknowledgments

Firstly, I would like to thank my supervisor, Gengwei Zhang, whose encouragement have pushed me to sharpen my thinking and given me financial support. Secondly, I would like to acknowledge my team members,Siyuan Li, Xijie Ke, Zhenyu Zhou, Lijun Zhang, Baisong

Yang, Bilei Ji, Liming Jiang, Ting Huang, without your friendly assistance and patient support, this study could not have been completed. Finally, I would like to acknowledge the great Shenzhen Third People's Hospitall, With the goal of becoming a world class hospital that leads the world, Shenzhen Third People's Hospital strives to better construct the National Clinical Research Center for Infectious Diseases and to become a National Comprehensive Regional Medical Center that integrates functions including medical service, medical education, research, disease prevention, and early warning, featuring world-leading medical excellence, advanced medical equipment and environment, refined medical services, well-regulated diagnosis and treatment, and smart management.

## Author Contributions

**Conceptualization:** Siyuan Li.

**Data curation:** Xijie Ke.

**Formal analysis:** Zhenyu Zhou.

**Investigation:** Lijun Zhang, Yang Yang.

**Methodology:** Baisong Yang, Bilei Ji.

**Project administration:** Liming Jiang.

**Validation:** Ting Huang.

**Writing – original draft:** Gengwei Zhang.

**Writing – review & editing:** Haifeng Chang.

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
