## [Decision Letter · Decision Letter 0]

15 Aug 2024

PONE-D-24-16756Differences in Emergency Hospitalization Trauma Patients During and After the COVID-19 PandemicPLOS ONE

Dear Dr. Zhang,

Thank you for submitting your manuscript to PLOS ONE. After careful consideration, we feel that it has merit but does not fully meet PLOS ONE’s publication criteria as it currently stands. Therefore, we invite you to submit a revised version of the manuscript that addresses the points raised during the review process.

**Dear Respectable Authors**

**We received the reviewer's comments and our decision is Major revision. **

**Please send your responses as soon as possible and point-by-point. Please highlight the changes in yellow.**

**Cheers**

We look forward to receiving your revised manuscript.

Kind regards,

Morteza Arab-Zozani, Ph. D.

Academic Editor

PLOS ONE

2. In the online submission form, you indicated that your data is available only on request from a third party. Please note that your Data Availability Statement is currently missing the name of the third party contact or institution / contact details for the third party, such as an email address or a link to where data requests can be made. Please update your statement with the missing information.

Reviewers' comments:

Reviewer's Responses to Questions

**Comments to the Author**

1. Is the manuscript technically sound, and do the data support the conclusions?

Reviewer #1: Yes

Reviewer #2: Yes

2. Has the statistical analysis been performed appropriately and rigorously? 

Reviewer #1: Yes

Reviewer #2: I Don't Know

3. Have the authors made all data underlying the findings in their manuscript fully available?

Reviewer #1: Yes

Reviewer #2: Yes

4. Is the manuscript presented in an intelligible fashion and written in standard English?

Reviewer #1: Yes

Reviewer #2: No

5. Review Comments to the Author

Reviewer #1: This is a scientific sound well written manuscript. The study included all trauma patients admitted through the emergency department from January 2020 to March 2024. Clinical data were collected, and a retrospective comparative analysis was performed on the characteristics of the two groups of trauma patients during and after the pandemic, including gender, age, average length of hospital stay, proportions of admissions to various departments, proportions of severe cases, proportions of surgical patients, and mortality rates, using statistical methods for analysis.

Reviewer #2: Dear authors

The article with the title " Differences in Emergency Hospitalization Trauma Patients During and After the COVID-19 Pandemic" is a very important and popular topic in the world. Although the article is well be written, it is recommended that the following points consider:

1. Grammatically, revise the all the text of manuscript.

2. Implication outcomes of study is not been mentioned.

3. Strengths and weaknesses of the study, in addition to limitation of implementation, are not been mentioned.

4. The magazine writing format is not been followed.

5. The managerial and political implications of the study are not stated.

6. PLOS authors have the option to publish the peer review history of their article (what does this mean?). If published, this will include your full peer review and any attached files.

Reviewer #1: No

Reviewer #2: No

---

## [Author Response · Author response to Decision Letter 0]

18 Nov 2024

Dear Morteza Arab-Zozani, Ph. D

I am very sorry for my late reply, as I went to a remote area to provide support where the network signal was very poor.

 Thank you for your letter and for the reviewers' comments concerning our manuscript entitled Differences in Emergency Hospitalization Trauma Patients (PONE-D-24-16756R1).Those comments are all valuable and very helpful for revising and improving our paper,as well as the important guiding significance to our researches. We have studied comments carefully and lave made corrections which we hope to meet with PLOS ONE's approval.We highlighted the changes in yellow. 

Response:

1 We have made every effort to modify the style of the manuscript to meet with PLOS ONE's style requirements, including the file naming.

1.1 We have used Bold type, 18pt font for all Level 1 headings.

1.2 We have used bold type for all the figure titles.

1.3 We have used Bold type, 16pt font for all Level 2 headings. 

1.4 We have revised the references of more than 6 authors.

1.5 We have used double-space for the entire manuscript.

1.6 Unfortunately, we cannot convert xlsx format to tif or eps format,despite trying many ways, it may be because we didn't choose the right tool or we didn't fully understand the requirements. so we look forward to some information from you.

1.7 We have added Supporting information in accordance with the requirements of PLOS ONE.

1.8 We changed the file namings, to meet with PLOS ONE's style requirements,

2 .In the online submission form, I indicated that my data is available only on request from a third party.At that time, we didn't fully understand.We have made the modification,now all data are fully available without restriction.

3 I had included a separate caption for each figure in our manuscript.Maybe we didn't fully understand the requirements. so we look forward to some information from you.

Thank you very much for your attention and time.Please do not hesitate to contact us if there are any question,I will reply as soon as possible.Thanks again to the reviewers and editors for your hard work!Best wishes to you! 

Yours sincerely 

Gengwei Zhang 

October 30, 2024

---

## [Editor Report · Decision Letter 1]

26 Nov 2024

Differences in Emergency Hospitalization Trauma Patients During and After the COVID-19 Pandemic

PONE-D-24-16756R1

Dear Dr. Zhang,

We’re pleased to inform you that your manuscript has been judged scientifically suitable for publication and will be formally accepted for publication once it meets all outstanding technical requirements.

Kind regards,

Morteza Arab-Zozani, Ph. D.

Academic Editor

PLOS ONE
---

## [Editor Report · Acceptance letter]

6 Dec 2024

PONE-D-24-16756R1 

PLOS ONE

Dear Dr. Zhang, 

I'm pleased to inform you that your manuscript has been deemed suitable for publication in PLOS ONE. Congratulations! Your manuscript is now being handed over to our production team.

Kind regards, 

on behalf of

Dr. Morteza Arab-Zozani 

Academic Editor

PLOS ONE